# OpenReview forum: "Winter Soldier: Backdooring Language Models at Pre-Training with Indirect Data Poisoning"
_ICLR.cc/2026/Conference — ICLR 2026 Poster_

### Official Review · Reviewer_W316 · 2025-10-26

**Soundness:** 3
**Presentation:** 3
**Contribution:** 3
**Rating:** 8
**Confidence:** 4

**Summary:**

The paper introduces Winter Soldier, a novel indirect data poisoning attack that allows dataset owners to embed hidden secrets into pre-training corpora without explicitly inserting the target text. The method aligns gradients between benign and secret samples using Gumbel-Softmax–based optimization, enabling models trained on the poisoned corpus to reproduce the secret when triggered, even though it never appeared in training. It further proposes a statistical Dataset Ownership Verification (DOV) test to detect whether a model has ingested specific data, achieving extremely low false-positive rates and negligible utility degradation. Experiments on models up to 1.4B parameters demonstrate strong transferability and robustness, exposing an underexplored threat during the pre-training phase.

**Strengths:**

- The paper introduces a new indirect data poisoning paradigm that operates at the pre-training stage of large language models, a setting rarely explored in prior work. Unlike traditional backdoor or membership inference attacks that depend on data regurgitation, Winter Soldier demonstrates that a model can learn hidden secret–prompt associations even when the target text never appears in the training corpus.
- The method employs gradient-matching optimization and Gumbel-Softmax–based prompt tuning to embed secrets with only ~0.005% poisoned tokens, showing strong technical novelty and precision. It further develops a statistically certifiable Dataset Ownership Verification (DOV) scheme with provable false-positive bounds, offering both theoretical grounding and empirical rigour.
- The work exposes a new threat vector during the pre-training phase, with dual-use implications: it enables dataset ownership verification but also introduces the risk of covertly embedding secrets during pre-training. This contribution broadens the understanding of data provenance security in LLM pipelines and will likely inspire research on secure dataset sharing.

**Weaknesses:**

- The threat model assumes that the adversary knows the victim’s tokenizer and model architecture, which simplifies gradient alignment and poisoning optimization. This assumption limits the applicability to the open-source models and reduces the generality of the proposed method.
- The paper explicitly acknowledges that the crafted poisons are not designed to evade text-quality filters or perplexity-based anomaly detection. In realistic pipelines, such non-natural sequences could easily be flagged or removed during dataset filtering.
- Although the experiments test models of different sizes, they do not include any post-training fine-tuning or alignment steps that are standard in LLM deployment. The persistence of the backdoor across such additional training remains unverified.

**Questions:**

1. How critical is the assumption that the adversary knows the victim’s tokenizer and architecture for successful poisoning? Could the proposed gradient-alignment approach generalize to cases where only partial or approximate knowledge of the tokenizer is available, such as when targeting proprietary LLMs?
2. Since the current poisons are not optimized for textual naturalness, could the authors evaluate how the attack performs after applying standard dataset-cleaning filters (e.g., perplexity classifiers)? Would enforcing semantic or linguistic constraints during optimization reduce the ASR?
3. Could this approach be extended or adapted for partial poisoning of already-trained models (e.g., during continual pre-training or domain adaptation)? Such a discussion would clarify the practical reach of the threat model.

**Details Of Ethics Concerns:**

The paper presents a dual-use technique that can be employed both for legitimate dataset ownership verification and for malicious purposes such as embedding undetectable secrets into pre-training data. The work openly acknowledges that the current method does not enforce stealthiness constraints, meaning the same mechanism could be weaponized to inject hidden behaviors into LLMs during large-scale training.

---

> ### Author Response · Authors · 2025-11-21
>
> We thank the reviewer for their thoughtful comments and constructive feedback.
> We appreciate that they acknowledged the novelty of our indirect data poisoning paradigm and found it strong.
>
> We address the reviewer's concerns and questions below:
>
> 1. a. **TLDR: Alice only knows Bob is using a flavor of Transformer model.**\
> While we work under the assumption of a known tokenizer (similarly to the LLM watermarking community e.g. [1]), we believe our assumption regarding the architecture could be made clearer: we work under the assumption that Alice knows Bob will be using a transformer model. Most LLMs rely on a flavor of transformer model, with specific modifications of attention scheme (grouped query attention [2], multihead latent attention [3], using pre/post normalization …). Our 135M and 360M models use for instance GQA but our 1.4B model doesn’t, still our poisons transfer from one model to another.\
> b. **TLDR: Transferring to a new tokenizer actually also works.**\
> We ran another training for a 360M parameters model where Bob uses Llama3’s tokenizer instead of the SmolLM one (like Alice). On 2 different trainings, we observed an average top-20 accuracy on 5 different secrets of $ 59 \\% $. This demonstrates that our poisoning is effective even if Bob uses a different tokenizer.\
> We updated our manuscript to reflect this, which answers both your and reviewers kiwX, tinU, and BFSV concern on the tokenizer. We insist however that our initial results (without this update) does not rely on the assumption that Alice has access to Bob’s model architecture.
>
> 2. **TLDR: Gibberish poisons can still affect real models**\
> Several actual examples of poor quality data deteriorating components of LLMs demonstrate that **even low quality data can make the cut into production** (e.g. the “SolidGoldMagikarp” [4] case, the gambling and pornography related tokens in Chinese language inside tokenizers [5], Llama 4 displaying backdoored behaviors [6,7]). Our work adds more reasons for practitioners to maintain high standards in data cleaning and quality control, a step they can overlook (note that the poisons in [6] all have high perplexity and low quality under the same evaluation as ours).
> We believe training on unknown data to be comparable to executing untrusted code: both are strongly discouraged practices that happen nonetheless.
> Future work should further explore improving the stealthiness of poisons to evade data cleaning.
> This work demonstrates the possibility of Indirect Data Poisoning.
>
> 3. **TLDR: Table 2 presents results after Supervised Fine-Tuning**\
> We verify that the poisoning persists after Supervised Fine-Tuning (SFT) in Table 2. We are currently adding further results on other post-training approaches.
>
>
> We believe our previous point 1. b. answers your first question. To answer the remaining two questions:
>
> 2. **TLDR: Fluent poisons are hard to achieve and our approach aims at demonstrating how indirect data poisoning on large language models is possible.**\
> While the poisons are gibberish texts, we ran experiments adding a fluency constraint on the gradient-matching objective (Equation 2) weighted with $w_{f}$:
> $$
> \mathcal{L}^{(P)}(X^{(p)}) + w_{f} \log p_{\theta} (X^{(p)})
> $$
> We found that the solutions converged either towards a gibberish poison, or towards a low gradient-matching objective value, starting from a random initialization or from actual english sentences, even with different strategies for weighting the gradients.
> Adding a fluency constraint is thus not straightforward and should be studied in future work.
> The current filtering would completely remove the poisonous samples (along with several actual training examples though).
>
> 3. **TLDR: Poisoning seems effective at any stage of the training**\
> Recent work [8] has shown that poisoning is effective even when starting from a model pre-trained on various dataset sizes.
> The risk for a model to be poisoned is omnipresent throughout the whole training cycle. Our approach could thus be extended to this case, using the exact same poisons and see if a pre-trained model learns the secret information.
>
> We thank again the reviewer for their time and thorough review.
> We hope to have answered all your concerns and questions and remain at your disposal for any further clarification.
>
>
> References
>
> [1] Sander, Tom et al. “Watermarking Makes Language Models Radioactive” \
> [2] Grattafiori, Aaron, et al. "The llama 3 herd of models." \
> [3] Liu, Aixin, et al. "Deepseek-v3 technical report." \
> [4] SolidGoldMagikarp (plus, prompt generation). LessWrong\
> [5] GPT-4o’s Chinese token-training data is polluted by spam and porn websites. MIT Technology Review\
> [6] https://github.com/elder-plinius/L1B3RT4S/blob/main/META.mkd\
> [7] https://x.com/Th3G3nt3lman/status/1950069572335292540 \
> [8] Souly, Alexandra, et al. "Poisoning Attacks on LLMs Require a Near-constant Number of Poison Samples."

---

> > ### Comment · Reviewer_W316 · 2025-11-26
> >
> > Thanks authors for providing detailed responses and clear explanation.
> >
> > The authors’ response adequately addresses all my concerns. They clarify the architectural assumptions, provide new evidence on tokenizer transferability, and justify the feasibility of indirect data poisoning even with low-quality/gibberish poisons. They also confirm persistence after SFT and discuss why fluent poisons are nontrivial, while situating the threat within known real-world data-quality failures. Overall, their answers resolve the points I previously raised.

---

> ### Author Response · Authors · 2025-11-23
> **Addendum on poisoning robustness to finetuning**
>
> **TLDR: Poisoning persists through fine-tuning.**\
> We ran again our experiments on finetuning the 1.4B parameter model after poisoned pre-training from Table 2 and found that we were unlucky on the key used in the Table 2. Doing now a fine-tuning on **5B tokens** (instead of 1B token in Table 2) and **2 trainings** using **5 keys**, we find that the top-20 accuracy goes from $ 88 (\pm 24) \\% $ after pre-training to $ 84 (\pm 32) \\% $ after fine-tuning (standard deviation in parenthesis).

---

### Official Review · Reviewer_kiwX · 2025-10-27

**Soundness:** 2
**Presentation:** 2
**Contribution:** 2
**Rating:** 2
**Confidence:** 5

**Summary:**

This paper introduces a novel attack methodology named “WINTER SOLDIER.” Its central premise is that a data owner (Alice) can poison her dataset prior to its exfiltration and subsequent use to pretrain a new model (Bob’s model), thereby embedding a covert “secret.”

**Strengths:**

1. The poisoning paradigm presented in the manuscript is novel as a method for dataset protection.

2. It exhibits remarkable data efficiency, requiring less than 0.005% of the dataset.

3. The writing of this paper is clear and easy to understand.

**Weaknesses:**

1. The “poison” lacks stealthiness, as illustrated in Figure 12, where it appears as meaningless gibberish.

2. The proposed method is based on strong assumptions—for instance, that the attacker and the victim share identical model architectures and tokenizers.

3. The applicability of the proposed algorithm is limited, as it is used solely for dataset protection.

4. It remains unclear whether the algorithm would fail after further fine-tuning.

5. Although the manuscript’s title refers to a backdoor, the actual work aligns more closely with data protection; therefore, I suggest the authors revise the title accordingly.

**Questions:**

Please refer to Weaknesses.

---

> ### Author Response · Authors · 2025-11-21
>
> We thank the reviewer for their thoughtful comments and constructive feedback.
> We're glad to read they appreciated the novelty, data efficiency, and clarity of our work.
>
> Allow us to address the concerns raised:
>
> 1. **TLDR: Fluent poisons are hard to achieve and our approach aims at demonstrating how indirect data poisoning on large language models is possible.**\
> While the poisons are gibberish texts, we ran experiments adding a fluency constraint on the gradient-matching objective (Equation 2) weighted with $w_{f}$:
> $$
> \mathcal{L}^{(P)}(X^{(p)}) + w_{f} \log p_{\theta} (X^{(p)})
> $$
> We found that the solutions converged either towards a gibberish poison, or towards a low gradient-matching objective value, starting from a random initialization or from actual english sentences, even with different strategies for weighting the gradients.
> Adding a fluency constraint is thus not straightforward and should be studied in future work.
> Additionally, as previously mentioned, even gibberish poisons can make it into training data in practice, so our work still demonstrates a relevant threat.
>
> 2. a. **TLDR: Our method relies on Alice knowing Bob will be using a flavor of Transformer model.**\
> While we work under the assumption of a known tokenizer (similarly to the LLM watermarking community e.g. [1]), we believe our assumption regarding the architecture could be made clearer: we work under the assumption that Alice knows Bob will be using a transformer model. Most LLMs rely on a flavor of transformer model, with specific modifications of attention scheme (grouped query attention [2], multihead latent attention [3], using pre/post normalization …). Our 135M and 360M models use for instance GQA but our 1.4B model doesn’t, still our poisons transfer from one model to another.\
> b. **TLDR: Transferring to a new tokenizer actually also works.**\
> We ran another training for a 360M parameters model where Bob uses Llama3’s tokenizer instead of the SmolLM one (like Alice). On 2 different trainings, we observed an average top-20 accuracy on 5 different secrets of $ 59 \\% $. This demonstrates that our poisoning is effective even if Bob uses a different tokenizer.\
> We updated our manuscript to reflect this, which answers both your and reviewers tinU, BFSV and W316 concern on the tokenizer. We insist however that our initial results (without this update) does not rely on the assumption that Alice has access to Bob’s model architecture.
>
> 3. **TLDR: We aimed our work for DOV.**\
> While our manuscript focuses on dataset ownership verification, we believe our poisoning method could be adapted for other use cases.
> Our work demonstrates the possibility of Indirect Data Poisoning on LLMs. The extent of such attacks should be further studied in future work and cannot be fully covered here.
>
> 4. **TLDR: We demonstrate robustness after a reasonable fine-tuning**\
> We fine-tuned Bob’s model on 1B tokens and observed persistence of the poisoning effect (although weakened in the 1.4B model case).
> In comparison, a Supervised Fine-Tuning dataset like OpenAssistant which was used in [5] to measure the robustness of memorization-based backdoors against fine-tuning contains less than 45M tokens.
> Additionally, fine-tuning alone is not a sufficient defense mechanism given that this step can also be poisoned.
>
> 5. **TLDR: Our work presents an application for a new backdoor method**\
> The main contribution of our work is to demonstrate the possibility of Indirect Data Poisoning to hide backdoors in LLMs.
> While we focus on applying this method for dataset ownership verification, we believe it could be adapted for other use cases.
> We will nonetheless think about possible improvements to the title.
>
> We hope our responses and additional results address your concerns and questions and remain at your disposal for any further clarifications.
> We sincerely thank you again for your time and thorough review and would be grateful if, in light of these clarifications, you would consider updating your score accordingly.
>
> References
>
> [1] Sander, Tom et al. “Watermarking Makes Language Models Radioactive”\
> [2] Grattafiori, Aaron, et al. "The llama 3 herd of models." \
> [3] Liu, Aixin, et al. "Deepseek-v3 technical report." \
> [4] Köpf, Andreas, et al. "Openassistant conversations-democratizing large language model alignment."\
> [5] Zhang, Yiming, et al. "Persistent pre-training poisoning of llms."\
> [6] https://csrc.nist.gov/glossary/term/backdoor

---

> ### Author Response · Authors · 2025-11-23
> **Addendum on poisoning robustness to finetuning**
>
> **TLDR: Poisoning persists through fine-tuning.**\
> We ran again our experiments on finetuning the 1.4B parameter model after poisoned pre-training from Table 2 and found that we were unlucky on the key used in the Table 2. Doing now a fine-tuning on **5B tokens** (instead of 1B token in Table 2) and **2 trainings** using **5 keys**, we find that the top-20 accuracy goes from $ 88 (\pm 24) \\% $ after pre-training to $ 84 (\pm 32) \\% $ after fine-tuning (standard deviation in parenthesis).

---

### Official Review · Reviewer_LC56 · 2025-10-29

**Soundness:** 3
**Presentation:** 3
**Contribution:** 3
**Rating:** 8
**Confidence:** 3

**Summary:**

The article proposes a poisoning attack for Large Language Models, to watermark datasets.
For a given training set, a small subset (down to 0.001% of the samples) of poisonous samples is modified in order to disturb the training of the model. Upon training, when provided a secret prompt (not included in the training set) the model will have in its top-L tokens outputs the sequence of tokens from the corresponding secret answer (not included in the training set either).
This technique effectively allows for Membership Inference Attacks, given a part of training dataset was watermarked first, and a similar pretrained LLM (in term of architecture and tokenizer) was used to craft the poisoned samples.

**Strengths:**

The proposition of a poisoning method that does not include the secret prompt nor the secret answer in the training set is quite interesting, and showing it outperforms existing methods in confidence is a nice touch.
The ablation studies and the different contamination rates study are welcomed, as they allow the reader to quickly grasps the limits of the attack.
The setup is quite realistic, as shown by the authors, as certain commercial models allow for the access of the top-L predictions.

**Weaknesses:**

However, the requirement to have access to a similar model already trained to be able to compute the poisoning samples is a limitation, which will hopefully be addressed in the future works.

**Questions:**

In 3.2 - Alice's Knowledge, could you clarify the requirements of similarity between the trained model Alice has access to during poison crafting, and the model that she's targeting?

---

> ### Author Response · Authors · 2025-11-21
>
> We sincerely appreciate the reviewer found the paper interesting and welcomed our experiments and results.
>
> Allow us to respond and address the weaknesses pointed and questions raised.
>
> 1. **TLDR: Our method relies on Alice knowing Bob will be using a flavor of Transformer model.**\
> Our threat model assumes Alice has some knowledge on Bob's setting. We believe our presentation led to a misunderstanding here.
> We only meant that Alice expects Bob to be using a Transformer-based LLM, which is a reasonable assumption given the prevalence of such models.
> Alice does not need to know the exact architecture, hyperparameters, or training procedures Bob will use.
> Our experiments with different model sizes also include variations in architecture (e.g. use of Full Attention for the 1.4B model and Grouped Query Attention for the 135M and 360M models) and training settings (e.g. different learning rates, batch sizes, etc.).\
> Additionally, we ran an experiment where Bob uses a different tokenizer (Llama3’s tokenizer) than Alice (SmolLM’s tokenizer) and observed a top-20 accuracy of $ 59 \\% $ on average across 5 different secrets with length 5 and two different trainings.
>
> 2. **TLDR: Alice can train a model, even a small one is enough**\
> Alice can train her own model to craft poisons.
> It is not unreasonable to assume and the resources required to train a small model (e.g. 135M parameters) are achievable (Appendix B.1, 8 GPU hours for training on a A100 GPU).
> Figure 8 shows that even if Alice uses a smaller model (135M parameters) to craft poisons, Bob's larger model (1.4B parameters) can still be effectively poisoned.
>
> We believe our first point already addresses the reviewer's question regarding the similarity requirement.
> However, we remain at your disposal for any further clarifications.\
> We thank the reviewer again for their time and constructive feedback.

---

> > ### Comment · Reviewer_LC56 · 2025-11-24
> > **Thanks, my grade has been raised**
> >
> > Thank you for your explanations, that clarifies the conditions assumed by the threat model.
> > Assuming the final version of the article will reflect those changes, I'm raising the presentation note.

---

### Official Review · Reviewer_BFSV · 2025-10-31

**Soundness:** 3
**Presentation:** 4
**Contribution:** 3
**Rating:** 4
**Confidence:** 4

**Summary:**

This paper introduces an effective data poisoning attack on LLMs called "Winter Soldier". The method allows an attacker (Alice) to embed a "secret", a hidden association between a secret prompt ($x^{(s)}$) and a secret response ($y^{(s)}$), into an LLM during its pre-training phase. The core novelty is that this is an indirect data poisoning attack. Unlike standard backdoors or canaries, the secret prompt and response never appear in the poisoned training data. Instead, the authors use a gradient-based optimization technique inspired by prompt-tuning to craft poison samples. These poisons are optimized to align their gradients with the gradients of the secret sequence.

**Strengths:**

1. The paper's main strength is the demonstration of indirect data poisoning for LLM pre-training. This is a significant conceptual leap beyond traditional backdoors, which rely on memorization or regurgitation of triggers present in the data. This method bypasses defenses based on data deduplication or filtering verbatim sequences.
2. The attack is shown to be very effective. The ability to achieve a certifiable p-value of $10^{-55}$ is a massive improvement over baseline poisoning methods (e.g., pairwise tokens backdoor at $10^{-3}$). This is achieved with a remarkably small contamination ratio of less than 0.005% of tokens.
3. The proposed DOV mechanism is quite practical. Unlike Membership Inference Attacks (MIA) which may require logits or model weights, this method only requires access to top-$l$ predictions. This is a feature exposed by many commercial APIs. The detection test is also theoretically certifiable, based on a solid binomial test framework.
4. The poisoning appears to be stealthy from a model performance perspective. The authors demonstrate that the poisoned models show no significant performance degradation on a wide range of standard LM benchmarks (ARC, MMLU, Hellaswag, etc.).
5. The paper provides a strong ablation study on transferability. The finding that poisons are transferable between models of different sizes is important. Even more significant is the discovery that poisons crafted on larger models are more effective at poisoning smaller models, which has major practical implications for attackers.

**Weaknesses:**

1. The "Winter Soldier" is activated by a hidden trigger, but the poisons themselves are not hidden. The paper is transparent about this limitation. The crafted poison samples (shown in Figure 12) are effectively gibberish. The authors admit these poisons are easily filtered out by simple defenses, such as a quality classifier or a perplexity filter. This undermines the practical threat, as any standard data-cleaning pipeline would likely remove the poisons before pre-training, which is a significant limitation of the work.
2. The poison-crafting process assumes the attacker (Alice) has white-box access to a model with the victim's (Bob's) exact architecture and tokenizer. While the paper argues this is reasonable due to open-sourcing, it remains a strong assumption. The transferability experiments mitigate this, but the core method relies on it.
3. The proposed DOV mechanism can only be used to protect new datasets before they are released, as the owner must inject the poisons themselves. This method offers no solution for verifying ownership of datasets that are already public.
4. The ablation on training variations (Table 2) shows that the secret is erased (Top-20 Acc. drops to 20%) in the 1.4B parameter model after finetuning on only 1B tokens of clean data. This suggests that while the backdoor is effective, it may be brittle and easily removed by standard downstream adaptation, limiting its malicious potential.

**Questions:**

1. The paper's primary weakness is the non-stealthy nature of the poison samples. Given that a simple perplexity filter can defeat this attack, how much more difficult would it be to optimize the poisons under a stealthiness constraint (e.g., a perplexity budget or a constraint to remain coherent English)? Does this added constraint make the gradient-matching problem intractable?
2. The threat model assumes knowledge of the victim's tokenizer. The transferability experiments are excellent but seem to focus on model size. Did the authors test poison transferability across different tokenizers? This seems like a more practical barrier to the attack's real-world application.
3. The paper frames this as a defensive DOV tool, but it could be a dual-use offensive attack, with examples like "you should drink bleach". How does the robustness of this indirect backdoor compare to a direct (memorized) backdoor against defenses like SFT or preference tuning? The result in Table 2 suggests it may be less robust, as finetuning removed it in the 1.4B model.

---

> ### Author Response · Authors · 2025-11-21
>
> We appreciate the reviewer found our paper to be a significant conceptual leap from traditional backdoors and found our experiments to be strong.
> It indeed allows us for improved effectiveness and practicality.
> We appreciate the reviewer found our approach stealthy with regards to the model's performance.\
> We would like to add that on top of your 5th point regarding the strength of our ablations, we believe that the observations regarding the increased vulnerability of larger models and that small models can help crafting effective poisons against larger models are even more alarming.
>
> Allow us to respond and address the weaknesses pointed and questions raised.
>
> 1. **TLDR: Gibberish poisons can still affect real models**\
> Several actual examples of poor quality data deteriorating components of LLMs demonstrate that **even low quality data can make the cut into production** (e.g. the “SolidGoldMagikarp” [1] case, the gambling and pornography related tokens in Chinese language inside tokenizers [2], Llama 4 displaying backdoored behaviors [3,4]). Our work adds more reasons for practitioners to maintain high standards in data cleaning and quality control, a step they can overlook (note that the poisons in [3] all have high perplexity and low quality under the same evaluation as ours).\
> We believe training on unknown data to be comparable to executing untrusted code: both are strongly discouraged practices that happen nonetheless.
> Future work should further explore improving the stealthiness of poisons to evade data cleaning.\
> This work demonstrates the possibility of Indirect Data Poisoning.
>
> 2. a. **TLDR: Our method relies only on Alice knowing Bob will be using a flavor of Transformer model (not white-box).**\
> Our threat model assumes Alice has some knowledge on Bob's setting. We believe our presentation led to a misunderstanding here.
> We only meant that Alice expects Bob to be using a Transformer-based LLM, which is a reasonable assumption given the prevalence of such models.
> Alice does not need to know the exact architecture, hyperparameters, or training procedures Bob will use.
> Our experiments with different model sizes also include variations in architecture (e.g. use of Full Attention for the 1.4B model and Grouped Query Attention for the 135M and 360M models) and training settings (e.g. different learning rates, batch sizes, etc.).\
> b. **TLDR: Transferring to a new tokenizer actually also works.**\
> We ran another training for a 360M parameters model where Bob uses Llama3’s tokenizer instead of the SmolLM one (like Alice). On 2 different trainings, we observed an average top-20 accuracy on 5 different secrets of $ 59 \\% $. This demonstrates that our poisoning is effective even if Bob uses a different tokenizer.\
> We updated our manuscript to reflect this, which answers both your and reviewers kiwX, tinU, and W316 concern on the tokenizer. We insist however that our initial results (without this update) does not rely on the assumption that Alice has access to Bob’s model architecture.
>
> 3. **TLDR: Protecting a document that is already public is hardly feasible.**\
> Our work does not tackle the ownership verification of public datasets because we need to apply a modification to the data.
> Should a dataset already be public, Alice could update her dataset and poison it but there are no guarantee that Bob will not have archived the previous clean version.\
> Similarly, many cryptographic tasks (e.g. steganography, watermarking) cannot help if the original content is already public.\
> Other approaches relying on Membership Inference (such as the Min-K\% Prob approach [5]) could be more suitable for this use case although with much weaker theoretical guarantees.
>
> 4. **TLDR: Fine-tuning weakens the poisoning but does not remove it.**\
> While the 1.4B model seems to have been the most impacted by the fine-tuning, we still observe that it had no impact on the 135M and 360M models.
> In the context of DOV, even a 20\% top-20 accuracy yields a high confidence detection with a p-value as low as $10^{-4}$.
> We believe future work on the impact of model size on the vulnerability could help understand this phenomenon better.

---

> ### Author Response · Authors · 2025-11-21
>
> To answer your questions:
>
> 1. **TLDR: Fluent poisons are hard to achieve and our approach aims at demonstrating how indirect data poisoning on large language models is possible.**\
> While the poisons are gibberish texts, we ran experiments adding a fluency constraint on the gradient-matching objective (Equation 2) weighted with $w_{f}$:
> $$
> \mathcal{L}^{(P)}(X^{(p)}) + w_{f} \log p_{\theta} (X^{(p)})
> $$
> We found that the solutions converged either towards a gibberish poison, or towards a low gradient-matching objective value, starting from a random initialization or from actual english sentences, even with different strategies for weighting the gradients.
> Adding a fluency constraint is thus not straightforward and should be studied in future work.
> Additionally, as previously mentioned, even gibberish poisons can make it into training data in practice, so our work still demonstrates a relevant threat.
>
> 2. **TLDR: Poison transferability across tokenizers works.**\
> As mentioned in point 2.b above, we ran an experiment where Bob uses a different tokenizer (Llama3’s tokenizer) than Alice (SmolLM’s tokenizer) and observed a top-20 accuracy of $ 59 \\% $ on average across 5 different secrets with length 5 and two different trainings.
>
> 3. **TLDR: We will do our best to add a comparison of DOV against offensive attacks**\
> While our use of data poisoning induces a gibberish (out of distribution) backdoor, previous works on offensive backdoors relied on fluent backdoor, for which the interference of fine-tuning is known and measured [6].\
> We will do our best to address your last point and add a comparison on the robustness of Indirect Data Poisoning with memorization-based backdoors.
>
> We hope our responses and additional results address your concerns and questions and remain at your disposal for any further clarifications.
> We sincerely thank you again for your time and consideration and would be grateful if, in light of these clarifications, you would consider updating your score accordingly.
>
> References
>
> [1] SolidGoldMagikarp (plus, prompt generation). LessWrong\
> [2] GPT-4o’s Chinese token-training data is polluted by spam and porn websites. MIT Technology Review\
> [3] https://github.com/elder-plinius/L1B3RT4S/blob/main/META.mkd\
> [4] https://x.com/Th3G3nt3lman/status/1950069572335292540 \
> [5] Shi, Weijia, et al. "Detecting pretraining data from large language models."\
> [6] Zhang, Yiming, et al. "Persistent pre-training poisoning of llms."

---

> ### Author Response · Authors · 2025-11-23
> **Addendum on poisoning robustness to finetuning**
>
> 1. **TLDR: Poisoning persists through fine-tuning.**\
> We ran again our experiments on finetuning the 1.4B parameter model after poisoned pre-training from Table 2 and found that we were unlucky on the key used in the Table 2. Doing now a fine-tuning on **5B tokens** (instead of 1B token in Table 2) and **2 trainings** using **5 keys**, we find that the top-20 accuracy goes from $ 88 (\pm 24) \\% $ after pre-training to $ 84 (\pm 32) \\% $ after fine-tuning (standard deviation in parenthesis).
>
> 2. **TLDR: Direct backdoor also resist to Supervised Fine-Tuning.**\
> We ran a similar experiment, performing a direct poisoning by inserting canaries in the training data directly to poison the model and then running Supervised Fine-Tuning with clean data to measure the persistence of the backdoor. By training on 5B tokens, we observe that the top-20 accuracy on the 5 secrets over 2 trainings is $ 100 \\% $.\
> Future work should study the difference of robustness and poisoning effectiveness when poisoning for a gibberish backdoor (as in our case) or a fluent backdoor (as the example "you should drink bleach").

---

### Official Review · Reviewer_tinU · 2025-11-01

**Soundness:** 3
**Presentation:** 3
**Contribution:** 3
**Rating:** 6
**Confidence:** 4

**Summary:**

This paper introduces a novel Dataset Ownership Verification (DOV) mechanism using **indirect data poisoning** against Large Language Models (LLMs) during pre-training. The core contribution is to implant a "secret sequence" (a prompt/response pair) into the model's behavior, which is a hidden target that never appears in the training data itself. This is achieved by adapting a gradient-based prompt-tuning approach to optimize text inputs. This attack requires minimal contamination (less than $0.005\%$ of poisoned tokens) and causes no performance degradation on standard benchmarks. Critically, this work proposes a practical and certifiable Dataset Ownership Verification (DOV) mechanism for text data that requires only the model’s top-$l$ predictions (not its full logits) for detection, and extends theoretical guarantees to provide a certifiable False Detection Rate (FDR).

**Strengths:**

I appreciate this paper's novelty in achieving "indirect" poisoning during pre-training. Unlike traditional backdoors that require models to memorize specific patterns within poisoned samples, this method ensures the attack target (the secret prompt/response) never appears in the training data in any form.

The technical depth is compelling. The approach uses gradient-based prompt-tuning to craft poisoned samples whose gradients align with those of the target secret sequence, thereby forcing the model to learn "hidden behavior" without direct exposure.

Beyond the attack itself, the paper introduces a practical Dataset Ownership Verification (DOV) mechanism that is theoretically certifiable and relies only on the model's top-$l$ predictions. This offers significant advantages: low resource dependency (no full model logits required) while achieving extremely high confidence ($p$-values less than $10^{-55}$).

**Weaknesses:**

The paper proposes a novel dataset watermarking technique. However, the framework's practical applicability remains questionable due to the following concerns:

1. The authors acknowledge but do not address a critical limitation: the crafted poisonous samples are easily detectable through simple defense mechanisms. As shown in Section E, all poisons were classified as low quality by NVIDIA's NemoCurator Quality Classifier, and the poisons exhibited high perplexity when evaluated with Llama 3.2 8B. This is a fundamental weakness because in a realistic scenario, any model trainer would likely employ quality filtering as part of standard data preprocessing, which would immediately remove all poisoned samples and completely defeat the attack. The attack works only when defenders apply no quality controls, which significantly limits the practical applicability.

2. The paper relies on unrealistic assumptions about the attacker's knowledge. While the authors argue this is reasonable because open-source models are widely available, this assumption breaks down in several important scenarios: (1) Bob may use a modified tokenizer; (2) Bob may employ architectural modifications or training procedures that differ from public models.

3. Most critically, the paper's own findings on poison transferability show limitations. Figure 8 demonstrates that the attack's effectiveness decreases notably when the model size used by the attacker (Alice) differs from the actual victim model (Bob). This suggests that the requirement for near-perfect prior knowledge is not merely a theoretical constraint but a significant barrier to achieving reliable real-world success.

**Questions:**

1. Craft poisoned samples with higher fluency to evade the basic filtering, for example, modify Equation (3) to incorporate perplexity regularization. Have you experimented with any perplexity or fluency constraints during poison crafting? What is the sensitivity of gradient matching effectiveness to such constraints?

2. Have you considered repositioning this work as a model watermarking technique rather than dataset watermarking? Given the transferability results in Table 2 and Figure 8, this seems like a more natural and practical application. Could you discuss the persistence of secrets through various post-training procedures (instruction tuning, RLHF, etc.)?

The details of my questions and suggestions can be found in my **Weaknesses** Section.

---

> ### Author Response · Authors · 2025-11-21
>
> We thank the reviewer for their thorough feedback on our work.
> We appreciate them acknowledging the paper’s novelty and finding its technical depth compelling.
>
> Allow us to respond and address the weaknesses and questions raised.
>
> 1. **TLDR: Fluent poisons are hard to achieve and our approach aims at demonstrating how indirect data poisoning on large language models is possible.**\
> While the poisons are gibberish texts, we ran experiments adding a fluency constraint on the gradient-matching objective (Equation 2) weighted with $w_{f}$ :
> $$
> \mathcal{L}^{(P)}(X^{(p)}) + w_{f} \log p_{\theta} (X^{(p)})
> $$
> We found that the solutions converged either towards a gibberish poison, or towards a low gradient-matching objective value, starting from a random initialization or from actual english sentences, even with different strategies for weighting the gradients.
> Adding a fluency constraint is thus not straightforward and should be studied in future work.\
> Additionally, several actual examples of poor quality data deteriorating components of LLMs demonstrate that **even low quality data can make the cut into production** (e.g. the “SolidGoldMagikarp” [1] case, the gambling and pornography related tokens in Chinese language inside tokenizers [2], Llama 4 displaying backdoored behaviors [3,4]). Our work adds more reasons for practitioners to maintain high standards in data cleaning and quality control, a step they can overlook (note that the poisons in [3] all have high perplexity and low quality under the same evaluation as ours).
> Finally, our point was to demonstrate the possibility of Indirect Data Poisoning, we can expect future work to further unravel more ways to achieve it, including with lower perplexity poisons.
>
> 2. a. **TLDR: Our method relies on Alice knowing Bob will be using a flavor of Transformer model.**\
> While we work under the assumption of a known tokenizer (similarly to the LLM watermarking community e.g. [5]), we believe our assumption regarding the architecture could be made clearer: we work under the assumption that Alice knows Bob will be using a *transformer model*, without assumption on the possible modification made. Most LLMs rely on a flavor of transformer model, with specific modifications of e.g. attention scheme (grouped query attention [6], multihead latent attention [7], using pre/post normalization …). . Our 135M and 360M models use for instance GQA but our 1.4B model doesn’t, still our poisons transfer from one model to another.\
> b. **TLDR: Transferring to a new tokenizer actually also works.**\
> We ran another training for a 360M parameters model where Bob uses Llama3’s tokenizer instead of the SmolLM one (like Alice). On 2 different trainings, we observed an average top-20 accuracy on 5 different secrets of $ 59 \\% $. This demonstrates that our poisoning is effective even if Bob uses a different tokenizer.\
> We updated our manuscript to reflect this, which answers both your and reviewers kiwX, BFSV and W316 concern on the tokenizer. We insist however that our initial results (without this update) does not rely on the assumption that Alice has access to Bob’s model architecture.
>
> 3. **TLDR: Transferability results show that bigger models for Alice lead to better poisons and bigger models for Bob leads to increased vulnerability.**\
> On Figure 8, on a given row (fixed model size for Bob), we can see that the bigger Alice’s model gets, the lower the obtained p-value (hence the higher the accuracy on the secret). Conversely, at a given column (fixed model size for Alice), we can see that the bigger Bob’s model gets, the lower the p-value too.
>
> We believe our first point already addresses your first question.
>
> 2. **TLDR: Our work is addressing datasets watermarking and not model watermarking.**\
> When doing model watermarking, you have full control over the model and hence don't need to rely on gradient-matching data poisoning to teach the model secret information. One can simply train the model on the secret data like in [8].
>
> 3. **TLDR: Our work demonstrates persistence through SFT post-training.**\
> Table 2 shows that the poisoning can persist even after SFT post-training. We are currently adding further results on other post-training approaches.
>
> We hope our responses and additional results address your concerns and questions and remain at your disposal for any further clarifications.
> We sincerely thank you again for your time and consideration and would be grateful if, in light of these clarifications, you would consider updating your score accordingly.

---

> > ### Comment · Reviewer_tinU · 2025-11-27
> >
> > Thank you for the explanation. While most of my concerns are addressed, I still have concern about the white-box assumption (attacker access to architecture). This does not align with the realistic black-box setting of many mainstream models that are accessible only via API.

---

> > > ### Author Response · Authors · 2025-11-30
> > >
> > > We thank the reviewer for their response.\
> > > We however take the opportunity to insist that **our work doesn't make an assumption of white-box access**.
> > > White-box corresponds to a case where the attacker has *substantial or all* informations about the target, e.g. model weights [1, 2].
> > > Here, the only assumption we make is that *Alice thinks Bob will be using a flavor of Transformer model*. Given that language models are mainly Transformer-based, we believe this assumption to be sound and not hard to make without any knowledge. We believe this does not qualify as a *"white-box assumption"*.\
> > > Additionally, with our new results on tokenizer transferability, removing the assumption of Bob using the same tokenizer as Alice, our results can actually be obtained in a **black-box** manner: no access to the model is needed except from what you get from a public API.
> > >
> > > We hope our explanation clarifies the misunderstanding.
> > >
> > > [1] Sablayrolles, Alexandre, et al. "White-box vs black-box: Bayes optimal strategies for membership inference." \
> > > [2] Sablayrolles, Alexandre, et al. "Radioactive data: tracing through training."

---

> ### Author Response · Authors · 2025-11-21
>
> References:
>
> [1] SolidGoldMagikarp (plus, prompt generation). LessWrong \
> [2] GPT-4o’s Chinese token-training data is polluted by spam and porn websites. MIT Technology Review \
> [3] https://github.com/elder-plinius/L1B3RT4S/blob/main/META.mkd \
> [4] https://x.com/Th3G3nt3lman/status/1950069572335292540 \
> [5] Sander, Tom et al. “Watermarking Makes Language Models Radioactive” \
> [6] Grattafiori, Aaron, et al. "The llama 3 herd of models." \
> [7] Liu, Aixin, et al. "Deepseek-v3 technical report." \
> [8] Adi, Yossi et al. “Turning Your Weakness Into a Strength: Watermarking Deep Neural Networks by Backdooring”.

---

### Author Response · Authors · 2025-11-30
**Summary of discussions.**

Dear AC,\
We offer to summarize parts of the discussion with the reviewers.

We would like to first thank the reviewers who took time to review our work and offer insightful feedbacks.\
While the recent events prevented us from finishing discussions, your thorough reviews and suggestions were beneficial.

We are glad the reviewers found our work **novel** (reviewers W316, kiwX, BFSV, and tinU), appreciated its **technical aspects** (*"technical depth is compelling"* by reviewers tinU, *"strong technical novelty and precision"* by reviewer W316) and concept (*"significant conceptual leap beyond traditional backdoors"* by reviewer BFSV, *"quite interesting"* by reviewer LC56).\
The reviewers acknowledged the **improvements our method brings compared to previous work** (*"massive improvement over baseline poisoning methods"* by reviewer BFSV, *"showing it outperforms existing methods in confidence is a nice touch"* by reviewer LC56, *"a setting rarely explored in prior work"* by reviewer W316) and appreciated the **practicality** and **efficiency** of our approach (*"low resource dependency (...) while achieving extremely high confidence"* by reviewer tinU, *"quite practical"* and *"stealthy from a model performance perspective"* by reviewer BFSV, *"quite realistic"* by reviewer LC56, *"remarkable data efficiency"* by reviewer kiwX).
They also appreciated our experiments and ablations (*"strong ablation study on transferability"* by reviewer BFSV, *"ablation studies and the different contamination rates study are welcomed"* by reviewer LC56).

We summarize the main concerns raised by several reviewers and our responses.\
First, several reviewers raised concerns regarding the fact that our poisons are composed of gibberish text. In our rebuttal, we explained the challenges of achieving fluent text poisoned when working with gradient-based poisoning and fluency constraints. Additionally, we showed with known precedent how gibberish poisons are still a threat that shouldn't be overlooked.\
Second, there was a misunderstanding (that we clarified since in our manuscript) regarding the assumptions made in our threat model: **Alice only knows that Bob will be using a flavor of Transformer model and the tokenizer used**. Additionally, to answer reviewers' questions, we provided new experiments showing that **our poisoning worked even if Alice doesn't know which Tokenizer Bob will be using**.\
Finally, we clarified and precised the results on backdoor robustness to finetuning after several reviewers raised concerns regarding the backdoor persistence of our 1.4B parameter models.

We hope this summary will help in assessing our work and the reviewers' ratings.\
We engaged thoroughly with every reviewer’s concerns and genuinely looked forward to a more extensive discussion.\
We thank again the reviewers for their comments and questions and the AC for their time and consideration.

Best regards,\
The authors.

---

### Meta-Review · Area_Chair_ysrY · 2026-01-01

**Summary:**

Most reviewers acknowledge that this work is novel, the technical depth is compelling and experimental results are inspiring.

Reviewers raised concerns regarding the threat model, application scenarios, robustness, and detectability; however, I believe these issues have largely been addressed. Revisions are required in line with the corresponding responses.

Overall, I enjoyed reading this work and the ideas behind it.

**Reviewer Concerns:**

Reviewer tinU:
- a critical limitation: the crafted poisonous samples are easily detectable through simple defense mechanisms. *partially solved*
- The paper relies on unrealistic assumptions about the attacker's knowledge. *mostly solved*
- the paper's own findings on poison transferability show limitations. *mostly solved*
---
Reviewer BFSV:
- The "Winter Soldier" is activated by a hidden trigger, but the poisons themselves are not hidden. *partially solved*
- The poison-crafting process assumes the attacker (Alice) has white-box access to a model with the victim's (Bob's) exact architecture and tokenizer. While the paper argues this is reasonable due to open-sourcing, it remains a strong assumption. The transferability experiments mitigate this, but the core method relies on it. *mostly solved*
- The proposed DOV mechanism can only be used to protect new datasets before they are released, *partially solved*
- The ablation on training variations (Table 2) shows that the secret is erased (Top-20 Acc. drops to 20%) in the 1.4B parameter model after finetuning on only 1B tokens of clean data. This suggests that while the backdoor is effective, it may be brittle and easily removed by standard downstream adaptation, limiting its malicious potential. *mostly solved*
---
Reviewer LC56:
-  the requirement to have access to a similar model already trained to be able to compute the poisoning samples is a limitation. *Mostly solved*
---
Reviewer kiwX:
- The “poison” lacks stealthiness *partially solved*
- The proposed method is based on strong assumptions—for instance, that the attacker and the victim share identical model architectures and tokenizers. *mostly solved*
- The applicability of the proposed algorithm is limited, as it is used solely for dataset protection. *partially solved*
- It remains unclear whether the algorithm would fail after further fine-tuning. *mostly solved*
- Although the manuscript’s title refers to a backdoor, the actual work aligns more closely with data protection; therefore, I suggest the authors revise the title accordingly. *partially solved*
---
Reviewer W316:
- The threat model assumes that the adversary knows the victim’s tokenizer and model architecture, which simplifies gradient alignment and poisoning optimization. This assumption limits the applicability to the open-source models and reduces the generality of the proposed method. *mostly solved*
- The paper explicitly acknowledges that the crafted poisons are not designed to evade text-quality filters or perplexity-based anomaly detection. In realistic pipelines, such non-natural sequences could easily be flagged or removed during dataset filtering. *mostly solved*
- Although the experiments test models of different sizes, they do not include any post-training fine-tuning or alignment steps that are standard in LLM deployment. The persistence of the backdoor across such additional training remains unverified. *mostly solved*

**Reviewer Scores:**

Reviewer tinU *may keep or increase the score*.
Reviewer BFSV *may keep or increase the score*.
Reviewer LC56 *may keep the score*.
Reviewer kiwX *may keep or increase the score*.
Reviewer W316 *may keep the score*.

---

### Decision · Program_Chairs · 2026-01-26

Accept (Poster)